# Characteristics of Kidney Transplant Recipients with Prolonged Pre-Transplant Dialysis Duration as Identified by Machine Learning Consensus Clustering: Pathway to Personalized Care

**DOI:** 10.3390/jpm13081273

**Published:** 2023-08-19

**Authors:** Charat Thongprayoon, Supawit Tangpanithandee, Caroline C. Jadlowiec, Shennen A. Mao, Michael A. Mao, Pradeep Vaitla, Prakrati C. Acharya, Napat Leeaphorn, Wisit Kaewput, Pattharawin Pattharanitima, Supawadee Suppadungsuk, Pajaree Krisanapan, Pitchaphon Nissaisorakarn, Matthew Cooper, Iasmina M. Craici, Wisit Cheungpasitporn

**Affiliations:** 1Division of Nephrology and Hypertension, Department of Medicine, Mayo Clinic, Rochester, MN 55905, USA; supawit_d@hotmail.com (S.T.); s.suppadungsuk@hotmail.com (S.S.); pajaree_fai@hotmail.com (P.K.); craici.iasmina@mayo.edu (I.M.C.); wcheungpasitporn@gmail.com (W.C.); 2Chakri Naruebodindra Medical Institute, Faculty of Medicine Ramathibodi Hospital, Mahidol University, Samut Prakan 10540, Thailand; 3Division of Nephrology, University of Mississippi Medical Center, Jackson, MS 39216, USA; jadlowiec.caroline@mayo.edu; 4Division of Transplant Surgery, Mayo Clinic, Phoenix, AZ 85054, USA; mao.shennen@mayo.edu; 5Division of Transplant Surgery, Mayo Clinic, Jacksonville, FL 32224, USA; mao.michael@mayo.edu; 6Division of Nephrology and Hypertension, Department of Medicine, Mayo Clinic, Jacksonville, FL 32224, USA; pvaitla@umc.edu; 7Division of Nephrology, Texas Tech Health Sciences Center El Paso, El Paso, TX 79905, USA; prakrati.c.acharya@gmail.com; 8Renal Transplant Program, University of Missouri-Kansas City School of Medicine/Saint Luke’s Health System, Kansas City, MO 64108, USA; napat.leeaphorn@gmail.com; 9Department of Military and Community Medicine, Phramongkutklao College of Medicine, Bangkok 10400, Thailand; wisitnephro@gmail.com; 10Division of Nephrology, Department of Internal Medicine, Faculty of Medicine Thammasat University, Pathum Thani 12120, Thailand; pattharawin@hotmail.com; 11Division of Nephrology, Department of Internal Medicine, Thammasat University Hospital, Pathum Thani 12120, Thailand; 12Deparment of Medicine, Division of Nephrology, Massachusetts General Hospital, Harvard Medical School, Boston, MA 02114, USA; pitch.nissa@gmail.com; 13Department of Surgery, Medical College of Wisconsin, Milwaukee, WI 53226, USA; macooper@mcw.edu

**Keywords:** kidney transplant, transplantation, prolonged pre-transplant dialysis, dialysis duration

## Abstract

Longer pre-transplant dialysis duration is known to be associated with worse post-transplant outcomes. Our study aimed to cluster kidney transplant recipients with prolonged dialysis duration before transplant using an unsupervised machine learning approach to better assess heterogeneity within this cohort. We performed consensus cluster analysis based on recipient-, donor-, and transplant-related characteristics in 5092 kidney transplant recipients who had been on dialysis ≥ 10 years prior to transplant in the OPTN/UNOS database from 2010 to 2019. We characterized each assigned cluster and compared the posttransplant outcomes. Overall, the majority of patients with ≥10 years of dialysis duration were black (52%) or Hispanic (25%), with only a small number (17.6%) being moderately sensitized. Within this cohort, three clinically distinct clusters were identified. Cluster 1 patients were younger, non-diabetic and non-sensitized, had a lower body mass index (BMI) and received a kidney transplant from younger donors. Cluster 2 recipients were older, unsensitized and had a higher BMI; they received kidney transplant from older donors. Cluster 3 recipients were more likely to be female with a higher PRA. Compared to cluster 1, cluster 2 had lower 5-year death-censored graft (HR 1.40; 95% CI 1.16–1.71) and patient survival (HR 2.98; 95% CI 2.43–3.68). Clusters 1 and 3 had comparable death-censored graft and patient survival. Unsupervised machine learning was used to characterize kidney transplant recipients with prolonged pre-transplant dialysis into three clinically distinct clusters with variable but good post-transplant outcomes. Despite a dialysis duration ≥ 10 years, excellent outcomes were observed in most recipients, including those with moderate sensitization. A disproportionate number of minority recipients were observed within this cohort, suggesting multifactorial delays in accessing kidney transplantation.

## 1. Introduction

End-stage kidney disease (ESKD) represents a significant global health challenge, imposing a considerable burden on both individuals and healthcare systems [1]. While kidney transplantation is widely recognized as the optimal modality due to its lower mortality, improved quality of life and reduced healthcare costs compared to hemodialysis (HD) or peritoneal dialysis (PD) [2,3,4,5], the number of patients on the waiting list for kidney transplantation consistently surpasses the availability level of kidney donors [6]. As of December 2020, there were 91,455 patients on the waitlist in the United States, whereas only 23,401 kidney transplants were performed in 2019 [7].

The life expectancy of ESKD patients on dialysis is generally poor. The United States Renal Data System (USRDS) reported that the overall death rate for ESKD patients between 2009 and 2019 was 19.7%, significantly higher than the 10.5% for kidney transplant recipients [6]. Each extra year of maintenance dialysis was correlated with a roughly 6% increase in mortality risk [8]. In addition, data from USRDS also found that the longer the duration of pretransplant dialysis, the lower the graft and patient survival after kidney transplantation [9,10]. Therefore, patients who had been on dialysis for more than 10 years before the transplant had a worse prognosis than those on dialysis for a shorter duration [11].

In particular, the survival rates after kidney transplantation in patients with prolonged pre-transplant dialysis warrant specific attention. Patients who underwent dialysis for more than 10 years prior to transplantation exhibited notably lower survival rates compared to those with shorter durations of dialysis [9]. For example, the one-year survival rates following transplantation stand at 97.3% for patients with shorter dialysis durations, while patients who have endured prolonged dialysis exceeding 10 years face a notably reduced one-year survival rate of 93.2% [10]. These disparities in survival rates underscore the imperative to address the formidable challenges encountered by patients enduring extended spans of dialysis prior to undergoing transplantation [12].

Even though research has focused on identifying recipients with long-term dialysis before transplant in order to reduce adverse outcomes, no machine learning (ML) approach has ever been utilized [11,13,14,15,16,17,18,19]. ML is a subfield of artificial intelligence (AI) that involves the development of algorithms and models that enable computers to learn from and make predictions or decisions based on data patterns. These methods empower the clinical decision-making process by analyzing the vast datasets present in electronic health records (EHR) to extract valuable insights and inform medical judgments [20,21,22,23,24,25]. A fundamental principle underlying machine learning is its ability to uncover hidden patterns of similarity and dissimilarity among a multitude of data variables, subsequently organizing these variables into coherent clusters that reveal meaningful associations [20,26]. Recent research has illuminated the potential of specific machine learning algorithms to outperform traditional analytical techniques, leading to heightened accuracy in tasks such as prediction and classification [27,28,29]. By harnessing the capabilities of machine learning, there is a prospect of identifying distinct clusters among patients with extended dialysis periods before transplantation, provided these algorithms can unveil the critical characteristics influencing graft and patient survival. This uncharted avenue of exploration, leveraging the power of machine learning, holds promise in enhancing our understanding of the complexities surrounding extended dialysis periods and their implications for kidney transplant recipients.

In this study, kidney transplant recipients who had been on dialysis for at least 10 years prior to transplant in the OPTN/UNOS database from 2010 to 2019 were characterized by consensus cluster analysis using an unsupervised machine learning approach, including a comparison of the outcomes of each cluster.

## 2. Materials and Methods

### 2.1. Data Source and Study Population

We used the Organ Procurement and Transplantation Network (OPTN)/United Network for Organ Sharing (UNOS) database to identify adult patients who received their first kidney-only transplant from 2010 to 2019 in the United States. We included patients who had been on dialysis for at least 10 years before their kidney transplant. We excluded (1) patients with prior kidney transplants, to avoid misclassifying their dialysis duration over multiple listings, and (2) multi-organ transplant patients, due to different organ allocation policies. The Mayo Clinic Institutional Review Board approved this study (IRB number 21-007698).

### 2.2. Data Collection

We abstracted the comprehensive list of recipient-, donor-, and transplant-related characteristics, as shown in Table 1, to include in cluster analysis. All variables had missing data ratios of less than 10%. We imputed missing data using the multivariable imputation by chained equation (MICE) method [30].

### 2.3. Clustering Analysis

An unsupervised ML was applied by conducting a consensus clustering approach to categorize clinical phenotypes of kidney transplant recipients with prolonged pre-transplant dialysis duration [31]. For this analysis, we utilized the “ConsensusClusterPlus” package, version 1.46.0, which is a publicly available software tool widely acknowledged for its ability to perform consensus clustering analysis [31]. The purpose of this approach was to identify distinct patient clusters based on relevant characteristics. A pre-specified subsampling parameter of 80% with 100 iterations and a number of potential clusters (k) ranging from 2 to 10 were used to avoid producing an excessive number of clusters that would not be clinically useful. The optimal number of clusters was determined by examining the consensus matrix (CM) heat map, cumulative distribution function (CDF), cluster-consensus plots with the within-cluster consensus scores and the proportion of ambiguously clustered pairs (PAC). The within-cluster consensus score, ranging between 0 and 1, was defined as the average consensus value for all pairs of individuals belonging to the same cluster [32]. A value closer to one indicates better cluster stability. PAC, ranging between 0 and 1, was calculated as the proportion of all sample pairs with consensus values falling within the predetermined boundaries [33]. A value closer to zero indicates better cluster stability [33]. The detailed consensus cluster algorithms used in this study are provided for reproducibility in the Appendix A.

### 2.4. Outcomes

Posttransplant outcomes included (1) death-censored graft failure and (2) patient death within 1 and 5 years. We defined death-censored graft failure as the need for dialysis or kidney retransplant, while censoring patients for death or at the last follow-up date reported to the OPTN/UNOS database.

### 2.5. Statistical Analysis

After consensus clustering analysis, we compared the characteristics and outcomes among the assigned clusters of patients with prolonged dialysis duration before kidney transplant. We tested the difference in clinical characteristics using analysis of variance (ANOVA) or the Kruskal–Wallis test, as appropriate, for continuous variables, and a Chi-squared test for categorical variables. We identified the key characteristics of each cluster using the standardized mean difference between each cluster and the overall cohort, with a pre-specified cut-off of >0.3. We demonstrated the risk of death-censored graft failure and patient death after kidney transplant using a Kaplan–Meier plot. We calculated the hazard ratio (HR) for death-censored graft failure and patient death using Cox proportional hazard analysis. We did not adjust the association of the assigned clusters with posttransplant outcomes for clinical characteristics because clinically distinct clusters were purposefully generated from the consensus clustering approach. We used R, version 4.0.3 (RStudio, Inc., Boston, MA, USA; http://www.rstudio.com/, accessed on 1 September 2022); ConsensusClusterPlus package (version 1.46.0) for consensus clustering analysis, and the MICE command in R for multivariable imputation by chained equation [30]. 

## 3. Results

### 3.1. Clinical Characteristics of Each Kidney Transplant Cluster

There were 158,367 kidney transplant recipients from 2010 to 2019 in the United States. Of these, 5092 (3.2%) had a prolonged dialysis duration of at least 10 years before receiving a kidney transplant. Therefore, we performed consensus clustering analysis on a total of 5092 kidney transplant patients. The mean age for this cohort was 50.5 years. Over half (52%) of recipients with prolonged dialysis were black and 25% were Hispanic. The median dialysis duration was 11.7 (IQR 10.7–13.9) years. A total of 88% of the recipients received kidney transplants from standard non-extended criteria donors (non-ECD).

Recipient characteristics associated with lower death-censored graft survival included male sex, black race, higher body mass index (BMI), peripheral vascular disease and positive HIV serostatus, whereas donor characteristics associated with lower death-censored graft survival included ECD deceased donor, older donor age, female sex, black race, hypertensive donor and Kidney Donor Profile Index (KDPI) ≥ 85. In addition, more HLA mismatch, increased cold ischemia time, and delayed graft function were associated with lower death-censored graft survival. Meanwhile, recipient characteristics associated with lower patient survival were older age, male sex, higher BMI, longer pre-transplant dialysis duration, history of diabetes, malignancy, peripheral vascular disease, low functional status and lower serum albumin, whereas donor characteristics associated with lower patient survival were older donor age, hypertensive donor and KDPI ≥ 85. Delayed graft function was associated with lower patient survival (Appendix A).

Figure 1A shows the CDF plot consensus distributions for each cluster of prolonged pre-transplant dialysis recipients; the delta area plot shows the relative change in the area under the CDF curve (Figure 1B). The largest changes in the area occurred with k = 3, at which point the relative increase in the area became noticeably smaller. As shown in the CM heat map (Figure 1C, Appendix A), the ML algorithm identified cluster 3 with clear boundaries, indicating good cluster stability over repeated iterations. The mean cluster consensus score was highest in cluster 3 (Figure 2A). In addition, favorable low PAC was demonstrated for three clusters (Figure 2B). Thus, using baseline variables at the time of transplant, the consensus clustering analysis identified three clusters that best represented the data pattern of kidney transplant recipients with prolonged pre-transplant dialysis.

Consensus clustering analysis identified three clinically distinct clusters. There were 2043 (40%) patients in cluster one, 2152 (42%) patients in cluster two, and 896 (18%) patients in cluster three. Table 1 shows recipient-, donor-, and transplant-related characteristics of prolonged pre-transplant dialysis patients according to the assigned clusters. Cluster 1 recipients were more likely to be male (67%) and to be younger in age (mean recipient age, 44 ± 9 years). They were likely to be black (46%) or Hispanic (32%). They were unlikely to be diabetic (5%, DM), had a lower BMI (mean 25.6 kg/m^2^) and were unlikely to be sensitized (panel reactive antibody, PRA, median 0%). Most recipients in this cluster (96%) received standard KDPI kidneys from younger (mean age 30.7 years) non-ECD donors (Table 1 and Figure 3).

Cluster 2 recipients were more likely to be male (66%) and older (mean recipient age 57 years). Most cluster 2 recipients were likely to be black (62%) or Hispanic (19%). They were more likely to be diabetic (38%) and had a higher BMI (mean 30.4 ± 5.8 kg/m^2^). Cluster 2 recipients were not sensitized (PRA, median 0%) and most (96%) were transplanted with standard KDPI kidneys from older (mean age 45 years) non-ECD donors (Table 1).

Cluster 3 recipients were more likely to be female (79%). The majority were black (52%) or Hispanic (25%). Recipients in cluster 3 were sensitized (PRA, median 85%). Most (92%) received kidney transplants from standard KDPI donors with a lower number of total HLA mismatches (median 4), and less use of non-depleting induction (basiliximab 9%). Cluster 3 recipients were more likely to receive a kidney through national allocation (22%).

### 3.2. Posttransplant Outcomes of Each Kidney Transplant Cluster

Table 2 shows posttransplant outcomes according to the assigned clusters. Cluster 2 recipients had lower patient survival compared to clusters 1 and 3 (Figure 4A). Death-censored graft survival was also inferior in cluster 2 recipients (Figure 4B). One-year patient survival was 98.2%, 93.2% and 97.3% for clusters 1, 2 and 3, respectively; one-year death censored graft survival was, respectively, 96.3%, 93.2% and 95.8% (Table 2).

## 4. Discussion

Prolonged dialysis duration preceding kidney transplantation has been linked to inferior transplant outcomes [9,10,11]. However, outcomes for kidney transplant recipients with a dialysis duration exceeding 10 years were, overall, excellent. Surprisingly, this cohort comprised a disproportionately high percentage of black (52%) and Hispanic (25%) individuals. Despite the older recipient age in cluster 2 and the moderate sensitization in cluster 3, the majority of recipients in the study received a standard kidney, likely due to their high allocation priority, a result of prolonged dialysis duration.

In our study, we utilized an unsupervised machine learning technique to classify patients with extended dialysis durations before transplantation. These patients were drawn from the OPTN/UNOS database and categorized into three distinct clusters. Cluster 1 was comprised of younger, non-diabetic recipients who obtained kidneys from comparatively youthful donors. Individuals in cluster 2 were relatively older with higher BMIs; however, they were typically non-diabetic and received kidneys with a standard KDPI, and procured from slightly older donors. The majority of recipients in cluster 3 were female individuals with an elevated PRA. Despite the fact that these discrete recipient clusters exhibited varying clinical outcomes, including graft longevity and patient survival, the overall results across all clusters were positive.

The proportion of black and Hispanic recipients having greater than 10 years of dialysis time highlights health care disparities for minorities and the need for improved access to kidney transplantation. This observed percentage of minorities is significantly higher as compared to the prevalent dialysis population [34]. Although the current OPTN dataset does not account for the time an individual remains inactive on the waiting list, it is likely that delays in referral to transplant are responsible for the prolonged dialysis duration across all three clusters.

While delays in referral to transplantation likely play a role for many recipients in all three clusters, recipients in cluster 3 might have experienced additional delays due to moderate sensitization and a scarcity of compatible match offers. This is despite the current allocation system assigning some extra priority to those with moderate sensitization. Arguably, most recipients in cluster 3 were not highly sensitized and would not be expected to have such prolonged waiting times based on PRA alone. It is possible that additional barriers related to an overall lower number of black and Hispanic donors and lack of compatible matches for ethnic minorities may have further compounded this delay [35,36,37,38,39]. These observations underscore yet another barrier that minorities encounter in the transplantation process.

This study has several limitations. Due to the nature of the national registry cohort, it is not possible to identify exact causes of graft loss and patient death. Comparative studies from centers able to provide more granular data would help to confirm these findings. In addition, data specific to patients listed but inactive on the waitlist remains unavailable. Similarly, data on qualifying time is unavailable for comparison. Consequently, it is assumed that delays in referral to transplantation were the primary factor contributing to these extended dialysis times. It is, however, possible that other factors played a role. Another potential limitation of our study is the exclusion of multi-organ transplant patients, including kidney–pancreas recipients, which might limit the broader applicability of our findings. Notably, cluster 2 recipients exhibited higher rates of diabetes, which might suggest that the consideration of pancreas transplantation could indeed have been a contributing factor. Future investigations addressing the outcomes and challenges specific to kidney–pancreas recipients could offer additional insights. In addition, there was a small amount of missing data which could have impacted our clustering results. To reduce the likelihood of bias, we used the multivariable imputation by chained equation (MICE) approach to replace the missing data, which generates plausible values for missing data while preserving the original variability and dataset distribution. While techniques like SMOTE have their merits, our priority was to handle missing data while maintaining the statistical integrity of the original dataset. 

To the best of our knowledge, this is the first machine learning approach specifically targeting kidney transplant recipients with extended dialysis durations preceding transplantation. The outcomes of this machine learning clustering allow for a better understanding of characteristics in recipients with prolonged dialysis vintage. Future studies are necessary to individualize pre- and post-transplant care for recipients with prolonged dialysis duration before transplant in order to improve their outcomes.

Leveraging the novel insights offered by this machine learning approach, future studies will assume a crucial role in enhancing the optimization of the kidney transplantation process for patients who have undergone prolonged dialysis before transplantation. These studies should delve deeper into exploring and quantifying the distinct barriers that minorities and sensitized patients encounter during the kidney transplantation process. This is especially important, as these populations have been identified as experiencing extended wait times and reduced match rates. This will require an intersectional approach that blends demographic analysis with a comprehensive grasp of medical, social and systemic barriers. In addition, strategies should be developed to increase the pool of compatible donors for minority and sensitized patients. In terms of future implications, the identification of these unique patient clusters could help reshape the protocols around kidney transplantation. By tailoring the transplantation process based on the distinct requirements of various patient clusters, transplant providers might be able to reduce patients’ dialysis duration, enhance graft survival, and lower mortality rates. This strategy, in conjunction with policy modifications designed to tackle the obstacles encountered by minority and sensitized patients, has the potential to eventually foster a more equitable access to transplantation and yield improved overall outcomes.

## 5. Conclusions

In this OPTN/UNOS database, using an unsupervised machine learning clustering approach, three clinically distinct clusters of kidney transplant recipients with prolonged dialysis duration prior to transplant were identified. Although different clinical outcomes, including mortality and graft survival, were observed between these transplant recipient clusters, most recipients had excellent outcomes. A disproportionate number of minority recipients were observed within this cohort, suggesting multifactorial delays in accessing transplants.

## Figures and Tables

**Figure 1 jpm-13-01273-f001:**
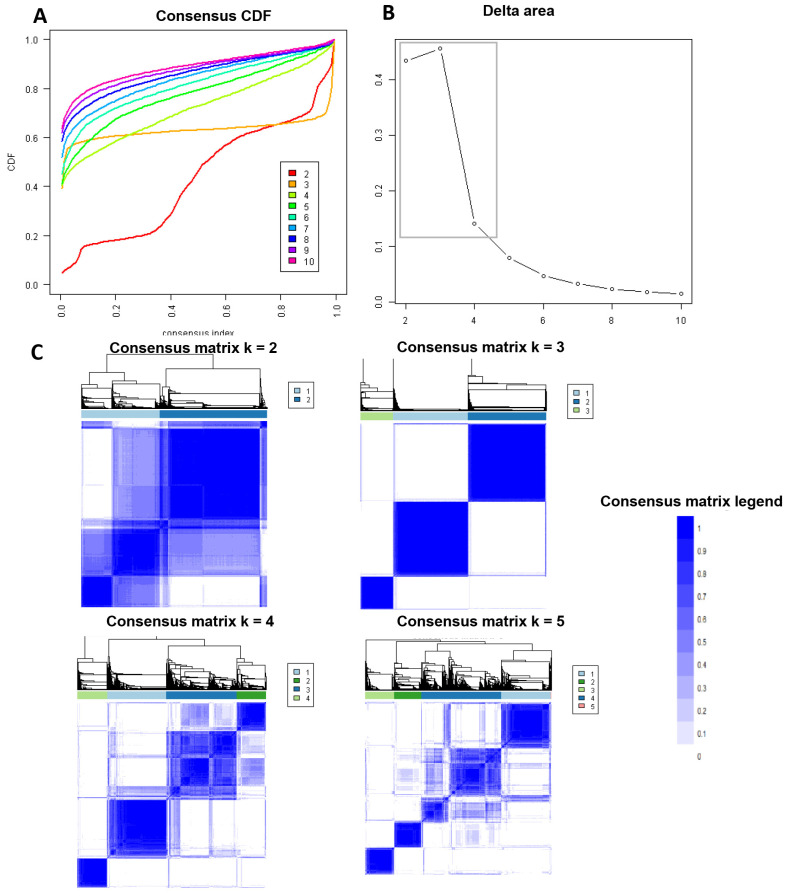
(**A**) CDF plot displaying consensus distributions for each k. (**B**) Delta area plot reflecting the relative changes in the area under the CDF curve. (**C**) Consensus matrix heat map depicting consensus values on a white to blue color scale of each cluster.

**Figure 2 jpm-13-01273-f002:**
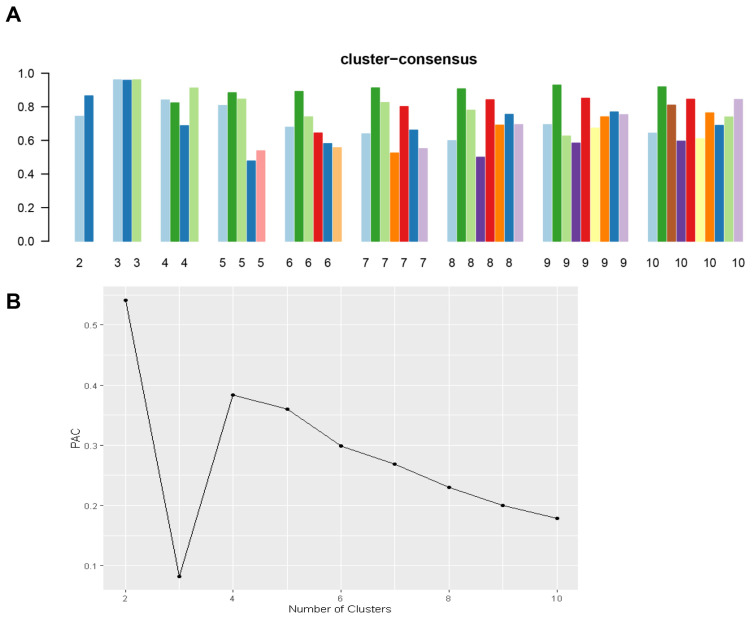
(**A**) The bar plot represents the mean consensus score for different numbers of clusters (k ranges from two to ten). Different colors indicate different cluster groups.; (**B**) The PAC values assess ambiguously clustered pairs.

**Figure 3 jpm-13-01273-f003:**
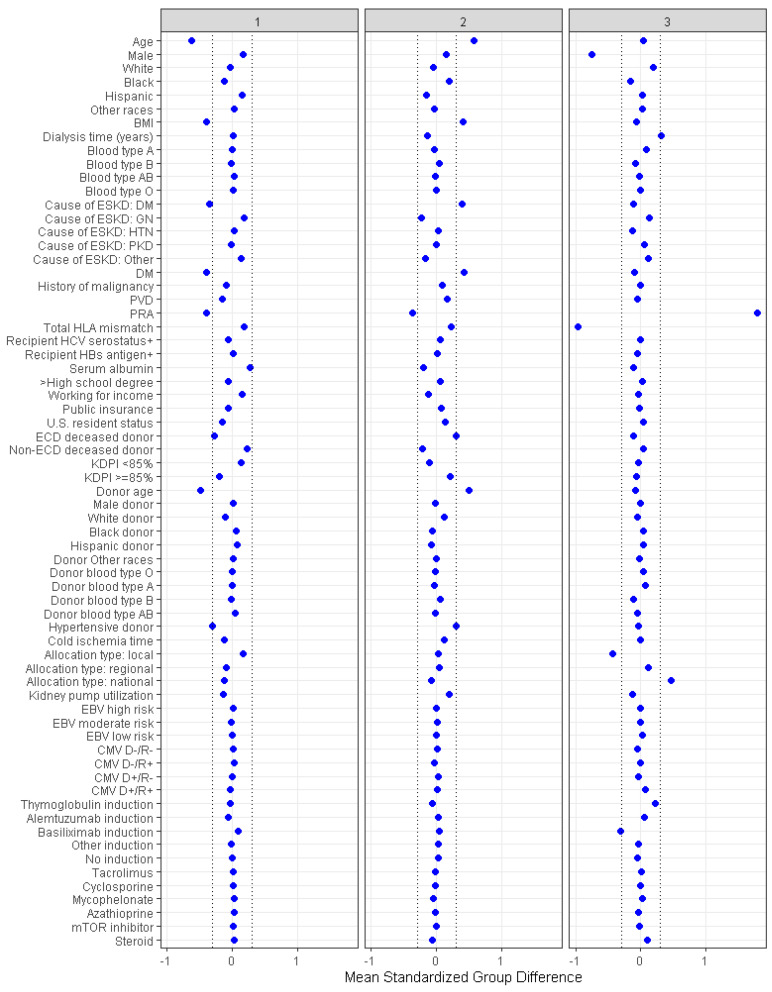
The baseline parameters’ standardized disparities across three distinct clusters are represented. The horizontal (x) axis indicates the value of standardized differences, whereas the vertical (y) axis depicts the baseline parameters. The dashed vertical lines serve as markers for the standardized difference cutoffs that are either less than −0.3 or greater than 0.3.

**Figure 4 jpm-13-01273-f004:**
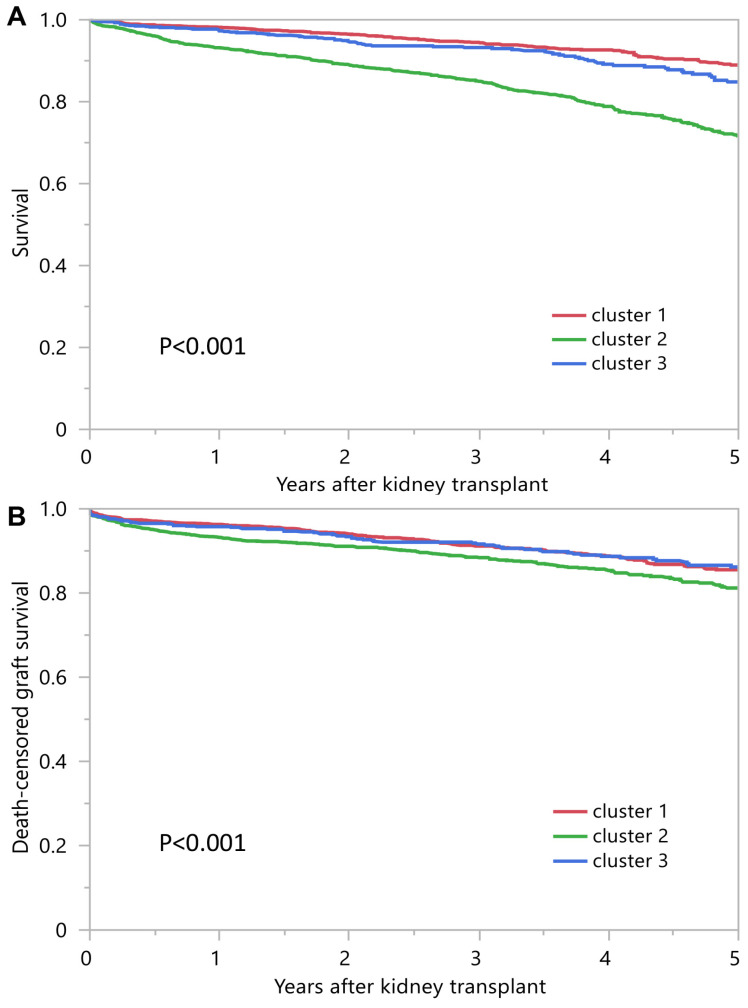
(**A**) Patient survival outcomes among three unique clusters of prolonged pretransplant dialysis recipients. (**B**) Death-censored graft survival.

**Table 1 jpm-13-01273-t001:** Clinical characteristics according to clusters of kidney transplant patients with long pretransplant dialysis duration.

	All (n = 5092)	Cluster 1 (n = 2043)	Cluster 2 (n = 2153)	Cluster 3 (n = 896)	*p*-Value
Recipient age (year)	50.5 ± 10.9	43.6 ± 9.3	56.9 ± 8.6	51.1 ± 10.1	<0.001
Recipient male sex	2990 (59)	1375 (67)	1426 (66)	192 (21)	<0.001
Recipient race					<0.001
- White	709 (14)	266 (13)	260 (12)	709 (14)	
- Black	2655 (52)	934 (46)	1324 (62)	2655 (52)	
- Hispanic	1286 (25)	649 (32)	403 (19)	1286 (25)	
- Other	442 (9)	194 (10)	166 (8)	442 (9)	
ABO blood group					0.007
- A	1109 (22)	438 (21)	445 (21)	226 (25)	
- B	856 (22)	336 (16)	398 (18)	122 (14)	
- AB	120 (17)	56 (3)	45 (2)	19 (2)	
- O	3007 (59)	1213 (59)	1265 (59)	529 (59)	
Body mass index (kg/m^2^)	28.0 ± 6.0	25.6 ± 5.0	30.4 ± 5.8	27.6 ± 6.0	<0.001
Dialysis duration (year), median (Q25, Q75)	11.8 (10.7–13.9)	11.9 (10.7–14.0)	11.5 (10.6–13.3)	12.4 (10.9–15.3)	<0.001
Cause of end-stage kidney disease					<0.001
- Diabetes mellitus	851 (17)	70 (3)	668 (31)	113 (13)	
- Hypertension	2111 (41)	878 (43)	917 (43)	316 (35)	
- Glomerular disease	1058 (21)	575 (28)	251 (12)	232 (26)	
- PKD	300 (6)	109 (5)	128 (6)	63 (7)	
- Other	772 (15)	411 (20)	189 (9)	172 (19)	
Comorbidity					
- Diabetes mellitus	1056 (21)	96 (5)	812 (38)	148 (17)	<0.001
- Malignancy	423 (8)	120 (6)	229 (11)	74 (8)	<0.001
- Peripheral vascular disease	535 (11)	123 (6)	332 (15)	80 (9)	<0.001
PRA, median (Q25, Q75)	0 (0,24)	0 (0,0)	0 (0,0)	85 (66,98)	<0.001
Positive HCV serostatus	393 (8)	124 (6)	203 (9)	66 (7)	<0.001
Positive HBs antigen	148 (3)	65 (3)	65 (3)	18 (2)	0.202
Positive HIV serostatus	262 (5)	136 (7)	107 (5)	19 (2)	<0.001
Functional status					<0.001
- 10–30%	13 (0)	2 (0)	8 (0)	3 (0)	
- 40–70%	2439 (48)	848 (42)	1144 (53)	447 (50)	
- 80–100%	2640 (52)	1193 (58)	1001 (46)	446 (50)	
Working income	886 (17)	472 (23)	272 (13)	142 (16)	<0.001
Public insurance	4661 (92)	1835 (90)	2011 (93)	815 (91)	<0.001
US resident	4858 (95)	1881 (92)	2114 (98)	863 (96)	<0.001
Undergraduate education or above	1985 (39)	728 (36)	899 (42)	358 (40)	<0.001
Serum albumin (g/dL)	4.0 ± 0.6	4.2 ± 0.6	3.9 ± 0.5	4.0 ± 0.6	0.335
Kidney donor status					<0.001
- Non-ECD deceased	4457 (88)	1937 (95)	1726 (80)	794 (89)	
- ECD deceased	456 (9)	25 (1)	379 (18)	52 (6)	
- Living	179 (4)	81 (4)	48 (2)	50 (6)	
Donor age	37.9 ± 14.5	30.7 ± 13.5	45.1 ± 11.7	36.7 ± 14.3	<0.001
Donor male sex	2998 (59)	1223 (60)	1248 (58)	527 (59)	0.458
Donor race					<0.001
- White	2871 (56)	1049 (51)	1343 (62)	479 (53)	
- Black	950 (19)	423 (21)	346 (16)	181 (20)	
- Hispanic	984 (19)	453 (22)	342 (16)	189 (21)	
- Other	287 (6)	118 (6)	122 (6)	47 (5)	
History of hypertension in donor	1275 (25)	243 (12)	822 (38)	210 (23)	<0.001
KDPI					<0.001
- Living donor	179 (4)	81 (4)	48 (2)	50 (6)	
- KDPI < 85	4707 (92)	1958 (96)	1928 (90)	821 (92)	
- KDPI ≥ 85	206 (4)	4 (0)	177 (8)	25 (3)	
HLA mismatch, median (Q25, Q75)	5 (4, 5)	5 (4, 6)	5 (4, 6)	4 (2, 5)	<0.001
Cold ischemia time (hours)	16.1 ± 8.8	15.1 ± 8.4	17.2 ± 9.2	16.1 ± 8.3	<0.001
Kidney on pump	2011 (39)	662 (32)	1048 (49)	301 (34)	<0.001
Delay graft function	1829 (36)	596 (29)	970 (45)	263 (29)	<0.001
Allocation type					<0.001
- Local	4197 (82)	1808 (89)	1797 (83)	592 (66)	
- Regional	449 (9)	125 (6)	215 (10)	109 (12)	
- National	446 (9)	110 (5)	141 (7)	195 (22)	
EBV status					0.888
- Low risk	26 (1)	10 (0)	10 (0)	6 (1)	
- Moderate risk	4626 (91)	1849 (91)	1963 (91)	814 (91)	
- High risk	440 (9)	184 (9)	180 (8)	76 (8)	
CMV status					0.088
- D−/R−	440 (9)	186 (9)	191 (9)	63 (7)	
- D−/R+	1427 (28)	599 (26)	577 (27)	251 (28)	
- D+/R+	2505 (49)	972 (48)	1064 (49)	469 (52)	
- D+/R−	720 (14)	286 (14)	321 (15)	113 (13)	
Induction immunosuppression					
- Thymoglobulin	2975 (58)	1161 (57)	1192 (55)	622 (69)	<0.001
- Alemtuzumab	714 (14)	247 (12)	324 (15)	143 (16)	0.004
- Basiliximab	1117 (22)	529 (26)	505 (23)	83 (9)	<0.001
- Other	83 (2)	29 (1)	43 (2)	11 (1)	0.194
- No induction	429 (8)	175 (9)	193 (9)	61 (7)	0.142
Maintenance Immunosuppression					
- Tacrolimus	4703 (92)	1897 (93)	1977 (92)	829 (93)	0.447
- Cyclosporine	76 (1)	35 (2)	29 (1)	12 (1)	0.568
- Mycophenolate	4806 (94)	1944 (95)	2010 (93)	852 (95)	0.025
- Azathioprine	7 (0)	5 (0)	2 (0)	0 (0)	0.196
- mTOR inhibitors	31 (1)	15 (1)	12 (1)	4 (0)	0.602
- Steroid	3783 (74)	1547 (76)	1534 (71)	702 (78)	<0.001

**Table 2 jpm-13-01273-t002:** Posttransplant outcomes according to the clusters.

	Cluster 1	Cluster 2	Cluster 3
1-year death-censored graft survival	96.3%	93.2%	95.8%
HR for 1-year death-censored graft failure	1 (ref)	1.84 (1.38–2.44)	1.16 (0.77–1.72)
5-year death-censored graft survival	85.5%	81.2%	86.2%
HR for 5-year death-censored graft failure	1 (ref)	1.40 (1.16–1.71)	1.00 (0.76–1.30)
1-year survival	98.2%	93.2%	97.3%
HR for 1-year death	1 (ref)	3.75 (2.61–5.52)	1.45 (0.84–2.46)
5-year survival	89.0%	71.5%	84.8%
HR for 5-year death	1 (ref)	2.98 (2.43–3.68)	1.38 (1.03–1.84)

## Data Availability

The data used in this study can be obtained upon reasonable request to the corresponding author.

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
