# Peer review of "Characteristics of Kidney Transplant Recipients with Prolonged Pre-Transplant Dialysis Duration as Identified by Machine Learning Consensus Clustering: Pathway to Personalized Care"

_jpm, 2023, doi:10.3390/jpm13081273_

Round 1

Reviewer 1 Report

Overall, the paper needs further English language refinements, as there are sentences without verbs in them, or the overall word topic could be improved.

The introduction could potentially benefit from the following suggestions:

- I suggest that the authors provide exact data and percentages for the lower survival rates after kidney transplantation in patients with prolonged pre-transplant dialysis .

- I would recommend that the authors elaborate more upon the definition of artificial intelligence, machine learning and computer-assisted diagnosis and decision support tools, as they represent the core of the paper.

For the Materials and methods section, I would like to ask the authors to clarify if the machine learning software was elaborated as an in-house algorithm or if an already-available program was used (and if so, please mention the name, manufacturer and version).

The Results need to highlight more clearly which variables were determined as being of negative prognostic value for patients' survival and graft function in the chosen cohort.

Finally, regarding the Discussion section, I would like to ask the authors to debate why other synthetic data augmentation methods were not considered (such as SMOTE) in order to "fill in" the missing variables, as well as commenting on the influence that this techniques might have on the final statistical soundness. 

The first sentence of the introduction doesn't make sense, as it lacks a verb.

Overall, the paper needs further English language refinements, as there are sentences without verbs in them, or the overall word topic could be improved.

Author Response

Response to Reviewer#1

Overall, the paper needs further English language refinements, as there are sentences without verbs in them, or the overall word topic could be improved.

Comment #1

Overall, the paper needs further English language refinements, as there are sentences without verbs in them, or the overall word topic could be improved.

Response: We appreciate your valuable feedback and thoughtful assessment of our paper. Your input is invaluable in enhancing the quality of our work. We have taken your comments into consideration and made the necessary revisions to address the concerns you raised regarding English language refinement and the overall coherence of the paper's topic. The following text has been revised comprehensively as suggested.

  1. Original Sentence: Longer dialysis duration predating kidney transplant has been associated with infe-rior transplant outcomes. Revised Sentence: Prolonged dialysis duration preceding kidney transplantation has been linked to inferior transplant outcomes.
  2. Original Sentence: Nonetheless, outcomes for kidney transplant recipients with >10 years dialysis duration were found to be overall excellent. Revised Sentence: However, outcomes for kidney transplant recipients with a dialysis duration exceeding 10 years were, overall, excellent.
  3. Original Sentence: This cohort somewhat sur-prisingly consisted of a disproportionately high percentage of Blacks (52%) and Hispanics (25%). Revised Sentence: Surprisingly, this cohort comprised a disproportionately high percentage of Black (52%) and Hispanic (25%) individuals.
  4. Original Sentence: The majority of the recipients in the study received a standard kidney despite older recipient age in cluster 2 and moderate sensitization in cluster 3, likely due to high alloca-tion priority for these individuals as a result of long dialysis duration. Revised Sentence: Despite the older recipient age in cluster 2 and the moderate sensitization in cluster 3, the majority of recipients in the study received a standard kidney, likely due to their high allocation priority resulting from prolonged dialysis duration.
  5. Original Sentence: In our study, we employed an unsupervised machine learning technique to classify patients with extended dialysis durations prior to transplant, drawn from the OPTN/UNOS database, into three unique clusters. Revised Sentence: In our study, we utilized an unsupervised machine learning technique to classify patients with extended dialysis durations before transplantation. These patients were drawn from the OPTN/UNOS database and categorized into three distinct clusters.
  6. Original Sentence: Individuals in Cluster 2 were somewhat older with a greater BMI, however, they were typically not diabetic and were given kidneys with a standard KDPI, sourced from slightly aged donors. Revised Sentence: Individuals in Cluster 2 were relatively older with higher BMIs; however, they were typically non-diabetic and received kidneys with a standard KDPI, procured from slightly older donors.
  7. Original Sentence: Most recipients in Cluster 3 were females with an elevated PRA. Revised Sentence: The majority of recipients in Cluster 3 were female individuals with an elevated PRA.
  8. Original Sentence: These observations highlight yet another barrier that minorities face in transplantation. Revised Sentence: These observations underscore yet another barrier that minorities encounter in the transplantation process.
  9. Original Sentence: Although delays in referral to transplant likely plays a role for many recipients in all three clusters recipients in cluster 3 may have experienced additional delays due moder-ate sensitization often and lack of compatible match offers despite the current allocation system assigning some additional priority to those with moderate sensitization. Revised Sentence: While delays in referral to transplantation likely play a role for many recipients in all three clusters, recipients in cluster 3 might have experienced additional delays due to moderate sensitization and a scarcity of compatible match offers. This is despite the current allocation system assigning some extra priority to those with moderate sensitization.
  10. Original Sentence: Similarly, data on qualifying time is not available for comparison. Revised Sentence: Similarly, data on qualifying time is unavailable for comparison.
  11. Original Sentence: As a result, it is assumed that delays in referral to transplant were primarily responsible for these prolonged dialysis times. Revised Sentence: Consequently, it is assumed that delays in referral to transplantation were the primary factor contributing to these extended dialysis times.
  12. Original Sentence: To our knowledge, this is the first machine learning approach specifically aimed at kidney transplant recipients with prolonged dialysis duration prior to transplant. Revised Sentence: To the best of our knowledge, this is the first machine learning approach specifically targeting kidney transplant recipients with extended dialysis durations preceding transplantation.
  13. Original Sentence: This study has several limitations. Revised Sentence: However, this study does have several limitations.
  14. Original Sentence: Building upon the novel insights provided by this machine learning approach, future studies will play a vital role in optimizing the kidney transplantation process for patients with prolonged dialysis duration before transplant. Revised Sentence: Leveraging the novel insights offered by this machine learning approach, future studies will assume a crucial role in enhancing the optimization of the kidney transplantation process for patients who have undergone prolonged dialysis before transplantation.
  15. Original Sentence: These studies should further explore and quantify the specific barriers that minorities and sensitized patients face in the kidney transplantation process, since these populations were identified as having longer wait times and lower match rates. Revised Sentence: These studies should delve deeper into exploring and quantifying the distinct barriers that minorities and sensitized patients encounter during the kidney transplantation process. This is especially important as these populations were identified as experiencing extended wait times and reduced match rates.
  16. Original Sentence: This will necessitate an intersectional approach, combining demographic analysis with a rigorous understanding of medical, social, and systemic barriers. Revised Sentence: This will require an intersectional approach that blends demographic analysis with a comprehensive grasp of medical, social, and systemic barriers.
  17. Original Sentence: By tai-loring the transplantation process according to the specific needs of different patient clus-ters, healthcare providers could potentially decrease the time patients spend on dialysis, improve graft survival, and decrease mortality rates. Revised Sentence: By tailoring the transplantation process based on the distinct requirements of various patient clusters, healthcare providers might be able to reduce patients' dialysis duration, enhance graft survival, and lower mortality rates.
  18. Original Sentence: This approach, coupled with policy changes aimed at addressing the barriers faced by minority and sensitized patients, could ultimately lead to more equitable access to transplantation and better overall outcomes. Revised Sentence: This strategy, in conjunction with policy modifications designed to tackle the obstacles encountered by minority and sensitized patients, has the potential to eventually foster a more equitable access to transplantation and yield improved overall outcomes.

Comment #2

The introduction could potentially benefit from the following suggestions:

- I suggest that the authors provide exact data and percentages for the lower survival rates after kidney transplantation in patients with prolonged pre-transplant dialysis .

Response: Following your suggestion, we have included precise data and percentages that underscore the lower survival rates after kidney transplantation in patients with prolonged pre-transplant dialysis. This addition serves to emphasize the critical nature of the problem we are addressing and provides readers with a clear quantitative context for the subsequent discussion.

“In particular, the survival rates after kidney transplantation in patients with pro-longed pre-transplant dialysis warrant specific attention. Patients who underwent dialy-sis for more than 10 years prior to transplantation exhibited notably lower survival rates compared to those with shorter durations of dialysis [9]. For example, the one-year sur-vival rates following transplantation stand at 97.3% for patients with shorter dialysis du-rations, while patients who have endured prolonged dialysis exceeding 10 years face a notably reduced one-year survival rate of 93.2% [10]. These disparities in survival rates underscore the imperative to address the formidable challenges encountered by patients enduring extended spans of dialysis prior to undergoing transplantation [12].”

- I would recommend that the authors elaborate more upon the definition of artificial intelligence, machine learning and computer-assisted diagnosis and decision support tools, as they represent the core of the paper

Response: We agree with the reviewer. We recognize the significance of thoroughly explaining the core concepts of artificial intelligence (AI), machine learning (ML), and computer-assisted diagnosis and decision support tools within our paper. In response to your recommendation, we have expanded the explanation of these concepts within the introduction. By doing so, we aim to ensure that readers without prior familiarity with these terms can readily comprehend their importance in the context of our research.

The following text has been added in the introduction as suggested.

Even though research has focused on identifying recipients with long-term dialysis before transplant in order to reduce adverse outcomes, no machine learning (ML) ap-proach has ever been utilized [11,13-19]. ML is a subfield of artificial intelligence (AI) that involves the development of algorithms and models that enable computers to learn from and make predictions or decisions based on data patterns [30]. These methods empower the clinical decision-making process by analyzing the vast datasets present in electronic health records (EHR) to extract valuable insights and inform medical judgments [20-25]. A fundamental principle underlying machine learning is its ability to uncover hidden patterns of similarity and dissimilarity among a multitude of data variables, subsequently organizing these variables into coherent clusters that reveal meaningful associations [20,26]. Recent research has illuminated the potential of specific machine learning algorithms to outperform traditional analytical techniques, leading to heightened accuracy in tasks such as prediction and classification [27-29]. By harnessing the capabilities of machine learning, there is a prospect of identifying distinct clusters among patients with extended dialysis periods before transplantation, provided these algorithms can unveil the critical characteristics influencing graft and patient survival. This uncharted avenue of exploration, leveraging the power of machine learning, holds promise in enhancing our understanding of the complexities surrounding extended dialysis periods and their implications for kidney transplant recipients.”

Comment #3

Materials and methods section, I would like to ask the authors to clarify if the machine learning software was elaborated as an in-house algorithm or if an already-available program was used (and if so, please mention the name, manufacturer and version).

Response: We appreciate and agree with the reviewer. In response to the reviewer's query regarding the machine learning software used in our study, we appreciate the opportunity to provide further clarification. The machine learning software utilized for our study was the "ConsensusClusterPlus" package, version 1.46.0, which is an established and publicly available tool. This package was employed to perform the consensus clustering analysis in order to categorize the clinical phenotypes of kidney transplant recipients with prolonged pre-transplant dialysis duration [31]. "ConsensusClusterPlus" is widely recognized for its effectiveness in conducting consensus clustering and is accessible within the R programming environment (R, version 4.0.3). We selected this package due to its established performance and its compatibility with the goals of our research. We additionally revised section of the "Materials and Methods" to include this clarification:

2.3. Clustering Analysis

An unsupervised ML was applied by conducting a consensus clustering approach to categorize clinical phenotypes of kidney transplant recipients with prolonged pre-transplant dialysis duration [31]. For this analysis, we utilized the "ConsensusClusterPlus" package, version 1.46.0, which is a publicly available software tool widely acknowledged for its ability to perform consensus clustering analysis [31]. The purpose of this approach was to identify distinct patient clusters based on relevant characteristics. A pre-specified subsampling parameter of 80% with 100 iterations and the number of potential clusters (k) ranging from 2 to 10 were used to avoid producing an excessive number of clusters that would not be clinically useful. The optimal number of clusters was determined by examining the consensus matrix (CM) heat map, cumulative distribution function (CDF), cluster-consensus plots with the within-cluster consensus scores, and the proportion of ambiguously clustered pairs (PAC). The within-cluster consensus score, ranging between 0 and 1, was defined as the average consensus value for all pairs of individuals belonging to the same cluster [32]. A value closer to one indicates better cluster stability. PAC, ranging between 0 and 1, was calculated as the proportion of all sample pairs with consensus values falling within the predetermined boundaries [33]. A value closer to zero indicates better cluster stability [33]. The detailed consensus cluster algorithms used in this study for reproducibility are provided in the Online Supplementary.

We trust that this clarification addresses the reviewer's query and provides a clear understanding of the machine learning software utilized in our study.

Comment #4

The Results need to highlight more clearly which variables were determined as being of negative prognostic value for patients' survival and graft function in the chosen cohort.

Response: We agree with the reviewer. We added Table S1 to demonstrate clinical characeristics associated with death-censored graft failrue and death in the cohort of kidney transplant recipients with prolonged pre-transplant dialysis duration.

The following statements have been added into the method.

Recipient characteristics associated with lower death-censored graft survival included male sex, black race, higher body mass index (BMI), peripheral vascular disease, positive HIV serostatus, whereas donor characteristics associated with lower death-censored graft survival included ECD deceased donor, older donor age, female sex, black race, hypertensive donor, Kidney Donor Profile Index (KDPI) ≥85. In addition, more HLA mismatch, increased cold ischemia time, and delayted graft function were associated with lower death-censored graft survival. Meanwhile, recipient characteristics associated with lower patient survival was older age, male sex, higher BMI, longer pre-transplant dialysis duration, history of diabetes, malignancy, peripheral vascular disease, low functional status, lower serum albumin, whereas donor characteristics associated lower patient survival was older donor age, hypertensive donor, KDPI ≥85. Delayed graft function was associated with lower patient survival (Table S1).

Comment #5

Finally, regarding the Discussion section, I would like to ask the authors to debate why other synthetic data augmentation methods were not considered (such as SMOTE) in order to "fill in" the missing variables, as well as commenting on the influence that this techniques might have on the final statistical soundness.

Response: We appreciate the reviewer's insightful comment regarding the use of synthetic data augmentation methods, particularly techniques like SMOTE, to address missing variables in our study. While these methods are indeed valuable for enhancing dataset completeness, we opted to utilize the multivariable imputation by chained equation (MICE) approach in our study due to its established effectiveness in handling missing data. MICE generates multiple imputations for missing data by creating plausible values based on observed relationships within the dataset, thereby maintaining the original variability and preserving statistical soundness.

In the context of our study, MICE was selected to maintain the integrity of the dataset while addressing missing values in a manner that is consistent with the characteristics of the original data. While techniques like SMOTE could potentially offer advantages in certain scenarios, such as generating synthetic data points for imbalanced datasets, our focus was primarily on imputing missing values in a way that aligns with the underlying data distribution.

We acknowledge the importance of discussing the potential influence of different data augmentation techniques on the final statistical soundness of the study. Below, we provide a revised version of the "Discussion" section that includes this consideration:

“In addition, there was a small amount of missing data which could have impacted our clustering results. To reduce the likelihood of bias, we used the multivariable imputation by chained equation (MICE) approach to replace the missing data, which generates plausible values for missing data while preserving the original variability and dataset distribution. While techniques like SMOTE have their merits, our priority was to handle missing data while maintaining the statistical integrity of the original dataset.”

Thank you for your time and consideration.  We greatly appreciated the reviewer's and editor's time and comments to improve our manuscript. The manuscript has been improved considerably by the suggested revisions.

Reviewer 2 Report

Generally, the English in the manuscript is excellent. There are a few examples where it could be improved:

Introduction:

  Although End-Stage Kidney Disease (ESKD) is one of the major worldwide health issues, placing a substantial burden on both patients and the healthcare system. 

  Although kidney transplantation is widely acknowledged as the best modality due to lower mortality, better quality of life and lower healthcare costs than hemodialysis (HD) or peritoneal dialysis (PD), the number of patients on the waiting list for kidney transplantation continues to exceed available kidney donors.

Results:

  Eighty-eight percent recipients received kidney transplants from standard non-extended criteria donors (non-ECD).

Discussion:

  Individuals in cluster 2 were somewhat older with a greater BMI, however, they were typically not diabetic and were given kidneys with a standard KDPI, sourced from slightly aged donors.

In the Materials and Methods section, the authors note that they included only patients who received kidney-only transplants from 2010 to 2019. They note that they excluded multi-organ transplant patients due to different organ allocation policies. I would suggest a comment specifically regarding kidney-pancreas recipients because (a) these patients are generally allocated organs in a manner aligned with kidney-only recipients and (b) Cluster 2 recipients had higher rates of diabetes so the double organ listing might have been a contributing factor.

In the Materials and Methods section, the authors note that they had to use logistic regression analysis because the date of rejection was not reported in the OPTN/UNOS database. Since this analysis could be distorted by inaccurate data or lack of available data, there should either be further comment regarding how much data that had to impute through logistic regression analysis or this analysis should be removed.

Generally, the English in the manuscript is excellent. There are a few examples where it could be improved:

Introduction:

  Although End-Stage Kidney Disease (ESKD) is one of the major worldwide health issues, placing a substantial burden on both patients and the healthcare system. 

  Although kidney transplantation is widely acknowledged as the best modality due to lower mortality, better quality of life and lower healthcare costs than hemodialysis (HD) or peritoneal dialysis (PD), the number of patients on the waiting list for kidney transplantation continues to exceed available kidney donors.

Results:

  Eighty-eight percent recipients received kidney transplants from standard non-extended criteria donors (non-ECD).

Discussion:

  Individuals in cluster 2 were somewhat older with a greater BMI, however, they were typically not diabetic and were given kidneys with a standard KDPI, sourced from slightly aged donors.

Author Response

Response to Reviewer#2

Comment #1

Generally, the English in the manuscript is excellent. There are a few examples where it could be improved:

Introduction:

 Although End-Stage Kidney Disease (ESKD) is one of the major worldwide health issues, placing a substantial burden on both patients and the healthcare system.

Response: We appreciate your thorough feedback on the manuscript's language quality. Thank you for pointing out the opportunity for improvement in the Introduction. We have revised the sentence as follows:

“End-stage kidney disease (ESKD) represents a significant global health challenge, imposing a considerable burden on both individuals and healthcare systems”

 Although kidney transplantation is widely acknowledged as the best modality due to lower mortality, better quality of life and lower healthcare costs than hemodialysis (HD) or peritoneal dialysis (PD), the number of patients on the waiting list for kidney transplantation continues to exceed available kidney donors.

Response: We sincerely appreciate your positive feedback regarding the manuscript's language quality. Your constructive input is valuable to us. We have revised the sentence you mentioned to enhance its clarity:

"While kidney transplantation is widely recognized as the optimal modality due to its lower mortality, improved quality of life, and reduced healthcare costs compared to hemodialysis (HD) or peritoneal dialysis (PD), the number of patients on the waiting list for kidney transplantation consistently surpasses the availability of kidney donors."

Results:

 Eighty-eight percent recipients received kidney transplants from standard non-extended criteria donors (non-ECD).

Response: We sincerely appreciate your positive feedback regarding the manuscript's language quality. Your feedback is invaluable to us. We have made the following revision to enhance the sentence:

"A total of 88% of the recipients received kidney transplants from standard non-extended criteria donors (non-ECD)."

Discussion:

 Individuals in cluster 2 were somewhat older with a greater BMI, however, they were typically not diabetic and were given kidneys with a standard KDPI, sourced from slightly aged donors.

Response: We are grateful for your positive assessment of the manuscript's language quality. Your input is truly appreciated. In response, we have revised the sentence for improved clarity and flow:

“Individuals in Cluster 2 were relatively older with higher BMIs; however, they were typically non-diabetic and received kidneys with a standard KDPI, procured from slightly older donors

Comment #2

In the Materials and Methods section, the authors note that they included only patients who received kidney-only transplants from 2010 to 2019. They note that they excluded multi-organ transplant patients due to different organ allocation policies. I would suggest a comment specifically regarding kidney-pancreas recipients because (a) these patients are generally allocated organs in a manner aligned with kidney-only recipients and (b) Cluster 2 recipients had higher rates of diabetes so the double organ listing might have been a contributing factor.

Response: We appreciate your insightful feedback and are grateful for your careful consideration of the study's methodology. We utilized the Organ Procurement and Transplantation Network (OPTN)/United Network for Organ Sharing (UNOS) database to identify adult patients who underwent their first kidney-only transplant within the United States between 2010 and 2019. Our inclusion criteria encompassed patients who had undergone dialysis for a minimum of 10 years prior to their kidney transplant. To ensure the integrity of our analysis, we excluded two categories of patients: 1) those with a history of prior kidney transplants to prevent potential misclassification of dialysis duration across multiple listings, and 2) recipients of multi-organ transplants due to variations in organ allocation policies that could confound our observations. It is noteworthy that our exclusion criteria were tailored to maintain the homogeneity of our study cohort.

However, we totally agree with the reviewer’s important point and we acknowledge the reviewer's suggestion regarding kidney-pancreas recipients. Although kidney-pancreas recipients share certain allocation policies with kidney-only recipients, we appreciate the potential impact of pancreas transplantation on our observations. Notably, Cluster 2 recipients exhibited higher rates of diabetes, which might suggest that the consideration of pancreas transplantation could indeed have been a contributing factor. We have included these important point in the discussion of our manuscript as suggested.

Comment #3

In the Materials and Methods section, the authors note that they had to use logistic regression analysis because the date of rejection was not reported in the OPTN/UNOS database. Since this analysis could be distorted by inaccurate data or lack of available data, there should either be further comment regarding how much data that had to impute through logistic regression analysis or this analysis should be removed.

Response: We agree with the reviewer and thus we deleted the analysis of rejection as suggested.

Thank you for your time and consideration.  We greatly appreciated the reviewer's and editor's time and comments to improve our manuscript. The manuscript has been improved considerably by the suggested revisions.

Round 2

Reviewer 1 Report

I congratulate the authors for the current form of the manuscript. I appreciate that the responded punctually to every suggestion. 

I consider the paper suitable for publication.